# It Is Time to Curb the Dogma in Lymphedema Management

Heather Barnhart [1,2]

1 Koya Medical, Oakland, CA 94607, USA; hbarnhart@koyamedical.com
2 Department of Physical Therapy, Nova Southeastern University, Fort Lauderdale, FL 33328, USA

**Abstract:** Lymphedema is an under-recognized and underappreciated disease. Advances in imaging and a deeper understanding of the pathophysiology of lymphedema are shedding new light on this disease that affects millions of people worldwide. As new evidence continues to emerge about the microcirculation and revised Starling Principle, etiological factors, related conditions, specific genes, and surgical innovations, the traditional approach to management must also evolve. This evolution is vital to maximize outcomes and improve quality of life. This commentary is a call to action to embrace innovation to better manage lymphedema and expand educational opportunities by leveraging technology to properly train healthcare providers to manage this disease.

**Keywords:** lymphedema; complete decongestive therapy

The "C" word. . .we have all experienced it, whether reluctantly or with arms wide open, as change is part of the human experience. So, why is change so difficult, especially in medicine? Change is difficult because we are comfortable with what we know and with entrenched dogma, such as "it's how it's always been done".

Professor Braithwaite at the Australian Institute of Health Innovation at Macquarie University perfectly summarizes the challenges of change in medicine [1]:

"For all the talk about quality healthcare, systems performance has frozen in time. Only 50–60% of care has been delivered in line with level 1 evidence or consensus based guidelines for at least a decade and a half [2–6]; around a third of medicine is waste, with no measurable effects or justification for the considerable expenditure [7–10]; and the rate of adverse events across healthcare has remained at about one in 10 patients for 25 years [11–14]. Dealing with this stagnation has proved remarkably difficult—so how do we tackle it in a new, effective way? We need to understand why system-wide progress has been so elusive and to identify the kinds of initiatives that have made positive contributions to date. Then we can ask what new solutions are emerging that may make a difference in the future and start to change our thinking about healthcare systems."

One area currently undergoing a renaissance is lymphatic medicine. Advances in imaging and a deeper understanding of the pathophysiology of the disease of lymphedema are shedding new light on this often under-recognized and underappreciated disease. As new evidence continues to emerge about the microcirculation and revised Starling Principle, etiological factors, related conditions, specific genes, and surgical innovations, the traditional approach to management must also evolve. This evolution is vital to maximize outcomes for the millions of individuals dealing with the disease of lymphedema.

Complete decongestive therapy (CDT) has been the mainstay for lymphedema management since the 1970s in Europe and the 1980s in the United States. CDTs roots are attributed to Alexander von Winiwarter (1848–1917), an Austrian surgeon who treated patients with limb swelling via elevation, compression, and massaging. Dr. Vodder (1896–1986) further manipulated the lymph nodes of his patients and subsequently developed 'lymph drainage massage'. Based on Vodder's work, Johannes Asdonk (1910–2003)

established the first school for manual lymph drainage (MLD) in 1969 in Germany. Additional work by Kuhnke, Foldi, Gregl, and others in the late 1970s established the German Society of Lymphology, which developed current CDT involving two phases of treatment (decongestion phase, followed by maintenance phase) encompassing skin care, manual lymphatic drainage, compression, and exercise [15].

Complete decongestive therapy has evidence to support its efficacy; however, there is room for innovation and refinement. Modern textiles are changing our compression options, advances in skin and wound care are improving integumentary integrity, imaging is helping providers to individualize MLD drainage patterns, and technologies are refining devices for home maintenance of lymphedema. Leveraging these advances enhances patient outcomes, thereby saving healthcare costs.

In the United States, the yearly charged inpatient costs for lymphedema exceed USD 1.3 billion, with 88% of admissions related to the lower extremity and 77% of patients directly admitted from the emergency department [16]. Forty-two percent of lower extremity lymphedema cases are caused by chronic venous insufficiency (CVI), and over 16 million individuals have phlebolymphedema (lymphedema of venous etiology) in the United States [17]. Adding to this issue, lymphedema is one of the most poorly understood, relatively underestimated, and least researched complications of cancer or its treatment [18]. In fact, more than 1 in 5 breast cancer survivors will develop breast cancer-related lymphedema (BCRL) [19]. Last but not least, lymphedema-associated cellulitis is estimated to generate USD 294.7 million in annual costs associated with hospitalization [16,20]. The number of patients diagnosed with lymphedema continues to rise, as do the associated costs required to manage these individuals. Compounding this issue are the various contributing factors that lead to poor outcomes. Underlying conditions and risk factors are on the rise (obesity, fat disorders, immobility, cancer survivorship, etc.) [21]; up to 18% of patients do not receive appropriate treatment [22]; less than 50% of patients adhere to their treatment regimen [23], due in part to the complexity of the maintenance phase [24]; and the need for and adoption of more innovative and novel interventions stymies outcomes [25]. This needs to change.

An article published in *Psychiatric Times* captured a dialogue and rebuttal about how hard changing healthcare practices can be and how changing culture is even harder. In the article, it is conveyed that in medicine, the only constant *is* change, and we must adapt to stay well-informed of the plethora of changes in technology, treatment, and medical knowledge. It goes on to say that effective and sustainable change in the healthcare system must be a partnership between policy makers, payers, and the people who deliver care day in, day out [26]. Patient-centered medicine must also partner with the patient to address their specific medical, physical, psychological, and cultural needs.

People do not inherently resist change; they resist *being* changed. With that in mind, it is vital that healthcare providers and policy makers stay relevant through education and training. This approach is the only way that successful paradigms can curb the dogma. Challenging tradition, asking why, and staying abreast of advances in research and medicine will support sustainable change in healthcare delivery. With respect to lymphedema, this approach means that it is ok to question traditional management approaches and embrace new preventative and management strategies supported by recent evidence. Current examples of paradigm shifts challenging traditional management models include using new technologies for the early detection of subclinical lymphedema, incorporating the use of lymphatic microsurgical techniques for lymphedema prophylaxis, and incorporating the use of exercise to help to manage lymphedema, which was traditionally considered to be unsafe, to name a few.

The PREVENT trial used bioimpedance spectroscopy (BIS) compared to tape measurements for the early detection and prevention of lymphedema in breast cancer patients [27]. The study demonstrated that BIS surveillance reduced rates of progression of breast cancer-related lymphedema (BCRL) by 10%, highlighting its ability to detect subclinical BCRL. Surgical innovations such as LYMPHA (lymphatic microsurgical preventative healing

approach) are helping to prevent lymphedema in patients who have undergone axillary lymph node dissection (ALND) [28]. Other surgical approaches, such as lymphatic bypass procedures, vascularized lymph node transfers, and suction assisted protein lipectomy, are showing promise in preventing and/or mitigating lymphedema in carefully selected patients. Further, evidence continues to support the use of exercise in patients with lymphedema to improve their range of motion, strength, fitness, and quality of life without exacerbating symptoms, which was a previously held belief [29].

Another promising shift in lymphedema identification and management is the use of ICG NIRFLI (indocyanine green near infrared fluoroscopy). This method provides real-time visualization of the lymphatic system to help physicians to plan surgical procedures and assist clinicians in customizing treatment interventions. By seeing and mapping the dysfunctional flow, clinicians can individualize manual lymphatic drainage pathways to functional areas (lymphosomes) specific to the patient. Rather than every patient receiving the same MLD treatment and pattern, this approach allows customization and improved outcomes by utilizing the patients' own anatomy and physiology [30].

Recognizing and leveraging exercise and mobility to enhance the muscle pump concomitantly improves venous and lymphatic return. Further, lymph nodes are strategically placed in anatomical areas that are constantly compressed, with range of motion, mobility, and breathing, encouraging lymphatic flow, cleansing, and fluid return to the venous system. Patients with lymphedema should be encouraged to be active within their clinical and medical presentation and incorporate management strategies that support mobility. The traditional approach of taking it easy and limiting activities is detrimental, given our current understanding of the interdependence of the venous, arterial, integument, and lymphatic systems [31].

Adapting manual techniques that rehabilitate the lymphatic and integumentary system should be accepted rather than challenged when shown to be efficacious. Siloed care is a disservice to patients. Specialists who treat patients with lymphedema should have sound knowledge of the lymphatic system, the vascular system, and the integumentary system to provide comprehensive management. Understanding lymphatic dermopathy is key to successful lymphedema management, as the body systems are interconnected.

Medical and allied health education is often taught through a body systems approach. Given the vast amount of knowledge that is necessary to master this field, the body systems are siloed out in most traditional didactic curricula. Oftentimes, the problem is that this siloed approach to learning leads to a siloed approach to management, neglecting the interplay and interconnections between the body systems. This issue is highlighted in lymphatic medicine when considering that the lymphatic system mediates immunity and inflammation and manages fluid. It is directly connected to the venous system and integumentary system and influences/is influenced by the endocrine, neuromuscular, cardiovascular, and GI systems. The successful management of patients with lymphedema must incorporate the integration of the body systems for comprehensive and holistic management. In consideration of this issue, lymphatic specialists are far outnumbered by patients with lymphatic disorders; therefore, new learning opportunities and educational delivery methods must adapt to provide accessible education, whether live, virtual, or blended.

The lymphatic, vascular, and integumentary communities must unite to advance science and challenge the dogma. Of course, more research is needed, but more importantly, we need to push boundaries and innovate. Patients should be empowered to manage their disease instead of their disease managing them. Let this paper be a call to action to embrace change and never stop asking why or how we can perform better. Let's collectively curb the dogma.

**Funding:** This article received no external funding.

**Conflicts of Interest:** Heather Barnhart is the Director of Clinical Affairs at Koya Medical and Faculty and Director of Wound Education at the International Lymphedema and Wound Training Institute

(ILWTI). She was a professor of physical therapy at Nova Southeastern University, where she remains an adjunct professor.

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
