# Peer review of "It Is Time to Curb the Dogma in Lymphedema Management"

_2813-3307, doi:10.3390/lymphatics1030016_

Round 1
Reviewer 1 Report
This is a very well-written commentary on an important topic. It could be improved and of greater benefit if the author would end the commentary with a few clear examples of specific proposed changes that would be meaningful and important to the management of the lymphedematous patient.
Author Response
Thank you for your kind review and suggestions. I have added additional supporting evidence and examples per your request. The new information is highlighted and can be found at lines 80-110, along with new references.
Reviewer 2 Report
It was a pleasure to review the manuscript It’s Time to Curb the Dogma in Lymphedema Management. Barnhart and colleagues present a commentary highlighting importance of embracing lymphedema, with a call to action to challenge historic non-evidence driven management. The focus of this manuscript is three-fold; 1- To highlight the importance of a significant disease process affecting patient quality of life and the health care system, 2- To excite new exploration in lymphedema evaluation and management, and 3- To highlight cultural resistance to change.
Comments for the authors:
Consider including and highlighting literature that challenge historic dogma.
- Literature that demonstrates a recent shift in secondary lymphedema (especially breast cancer related lymphedema) from traditional impairment models to prospective screening models. Such as the recent PREVENT Trial. Along with recent literature on the development of prospective screening programs.
- Literature supporting institution of novel surgical techniques for prevention such as immediate lymphatic reconstruction surgeries for lymphedema prevention (LYMPHA – lymphatic microsurgical healing approach)
Overall, I enjoyed reading this commentary and believe that it adds a new light to the literature regarding a significant health care problem. I believe this manuscript demonstrates passion to advance understanding into lymphedema and challenge established non-evidence-based norms.
Author Response
Thank you kindly for your supportive review. I have added content per your suggestions which is highlighted in lines 80-110, along with new references.
Most appreciated!